# Pediatric Respiratory Hospitalizations in the Pre-COVID-19 Era: The Contribution of Viral Pathogens and Comorbidities to Clinical Outcomes, Valencia, Spain

**DOI:** 10.3390/v16101519

**Published:** 2024-09-25

**Authors:** Valérie Bosch Castells, Ainara Mira-Iglesias, Francisco Xavier López-Labrador, Beatriz Mengual-Chuliá, Mario Carballido-Fernández, Miguel Tortajada-Girbés, Joan Mollar-Maseres, Joan Puig-Barberà, Javier Díez-Domingo, Sandra S. Chaves

**Affiliations:** 1New Products and Innovation Medical Franchise, Sanofi Vaccines, 69007 Lyon, France; sandra.chaves@sanofi.com; 2Área de Investigación en Vacunas, Fundación para el Fomento de la Investigación Sanitaria y Biomédica de la Comunitat Valenciana (FISABIO-Public Health), 46020 Valencia, Spain; ainara.mira@fisabio.es (A.M.-I.); jpuigb55@gmail.com (J.P.-B.); javier.diez@fisabio.es (J.D.-D.); 3Consorcio de Investigación Biomédica de Epidemiología y Salud Pública (CIBERESP), Instituto de Salud Carlos III, 28029 Madrid, Spain; f.xavier.lopez@uv.es (F.X.L.-L.); mengual_bea@gva.es (B.M.-C.); 4Área de Genómica y Salud, Fundación para el Fomento de la Investigación Sanitaria y Biomédica de la Comunitat Valenciana (FISABIO-Public Health), 46020 Valencia, Spain; 5Hospital General Universitario de Castellón, 12004 Castellón, Spain; carballido_mar@gva.es; 6Departamento de Medicina, Universidad CEU Cardenal Herrera, 12004 Castellón, Spain; 7Department of Pediatrics, Hospital Universitario Doctor Peset, 46017 Valencia, Spain; tortajadamig@gmail.com; 8Hospital Universitario y Politécnico La Fe, 46020 Valencia, Spain; mollar_jua@gva.es

**Keywords:** respiratory viruses, hospitalizations, children, comorbidities, complications, severity, viral codetection

## Abstract

Viral respiratory diseases place a heavy burden on the healthcare system, with children making up a significant portion of related hospitalizations. While comorbidities increase the risk of complications and poor outcomes, many hospitalized children lack clear risk factors. As new vaccines for respiratory viral diseases emerge, this study examined pediatric respiratory hospitalizations, focusing on viral etiology, complication rates, and the impact of comorbidities to guide future policy. Data were analyzed from eight pre-COVID influenza seasons (2011/2012–2018/2019) involving patients under 18 years hospitalized with respiratory complaints across 4–10 hospitals in Valencia, Spain. Respiratory specimens were tested for eight viral targets using multiplex real-time reverse-transcription polymerase chain reaction. Demographics, clinical outcomes, discharge diagnoses, and laboratory results were examined. Among the hospitalized children, 26% had at least one comorbidity. These children had higher rates of pneumonia, asthma exacerbation, and pneumothorax, and were twice as likely to require ICU admission, though mechanical ventilation and length of stay were similar to those without comorbidities. Respiratory syncytial virus (RSV) was the most common virus detected (23.1%), followed by rhinovirus/enterovirus (9.5%) and influenza (7.2%). Viral codetection decreased with age, occurring in 4.6% of cases. Comorbidities increase the risk of complications in pediatric respiratory illnesses, however, healthcare utilization is driven largely by otherwise healthy children. Pediatric viral vaccines could reduce this burden and should be further evaluated.

## 1. Introduction

Acute respiratory illnesses are an important cause of pediatric morbidity and mortality, with pneumonia being the main cause of death among children <5 years worldwide [1]. Viruses have been identified as the most common etiology of pneumonia in children [2], with those with comorbidities being at highest risk for complications [3]. For this reason, public health officials and policymakers have often prioritized people with comorbidities in health interventions and prevention measures [4,5,6,7]. Nonetheless, data among children with influenza, RSV, and more recently COVID-19, indicate that many children hospitalized with respiratory illness associated with these pathogens are otherwise healthy [5,8,9,10,11,12,13,14], contributing nonetheless to considerable healthcare utilization [15]. With the success of mRNA and adenovirus vaccine platforms [16], new vaccines against respiratory illnesses combining various viral targets have been discussed or are under development [17,18]. Young children, for instance, could potentially benefit from vaccines that target viruses, leading to medically attended illness. Investigating the impact of chronic medical conditions in children hospitalized with respiratory illness can improve our understanding of associated clinical outcomes and provide an evidence base to support policymakers on the prioritization of interventions.

We used data from a prospective active surveillance study conducted before the COVID-19 pandemic to investigate the characteristics of pediatric patients hospitalized with respiratory illness in healthcare district reference hospitals in the Valencia region, Spain. We describe the frequency of severe outcomes and complications by presence of comorbidities and explore associations with viral etiologies.

## 2. Materials and Methods

### 2.1. Population

We analyzed the data of patients aged <18 years hospitalized with respiratory complaints during the respiratory seasons 2011/12 to 2018/19. The respiratory season was defined as November to March/April for most seasons, except in 2017/18 where surveillance was extended from September to June and in 2018/19 where surveillance was year-round. This analysis used data collected from 4–10 hospitals (covering between 21% and 46% of the total population of the Valencia region of Spain, which is around 5 million) from a prospective hospital-based active surveillance study for acute respiratory illness [19,20,21,22,23]. Data collected during the 2019/20 and 2020/21 seasons were not included in our analysis as the preventive measures implemented during the COVID-19 pandemic impacted both healthcare seeking behavior and the circulation of respiratory viruses [24,25].

### 2.2. Inclusion Criteria

Children hospitalized with an acute respiratory illness, with an onset of <7 days from admission and resident in one of the participating hospitals’ catchment areas, were eligible for enrolling [26]. Patients discharged 30 days prior to the present admission were excluded to remove multiple enrolments during what have might been the same illness episode and to avoid the inclusion of nosocomial infections. Institutionalized patients and those who were not local residents were also excluded from the study.

### 2.3. Laboratory Methods

Enrolled patients had oropharyngeal and nasopharyngeal swabs if aged ≥14 years, or nasal and oropharyngeal swabs if aged <14 years, collected within 8–48 h of admission. Both swabs were combined in one tube of viral transport media (Copan, Brescia, Italy) and kept frozen at −50° to −20 °C until shipped refrigerated to a centralized virology laboratory at Fisabio-Public Health. One third of the viral transport media volume was used for the extraction of total nucleic acids using an automated silica-based method (Nuclisens e-Mag, BioMérieux, Lyon, France). Extracted nucleic acids (50 µL) were tested by five in-house multiplex real-time reverse transcription polymerase chain reaction (RT-PCR) screening assays for influenza, RSV, rhinovirus/enterovirus, adenovirus, coronavirus, bocavirus, metapneumovirus, and parainfluenza. Laboratory procedures to prevent RT-PCR contamination were strictly followed, and positive (purified viral nucleic acids Vircell, Granada, Spain) and negative controls (without sample and/or nucleic acid) were included. Positive results were defined, overall, by cycle threshold values up to 35 together with negativity of the negative controls. Negative results for viruses were only considered if the human ribonucleoprotein gene amplification was positive.

### 2.4. Data Sources and Categorization

Data were collected by interviewing parents/legal tutors and from the patients’ medical record on demographic characteristics, the presence of comorbidities, clinical course, and outcomes (including admission to the intensive care unit (ICU), need for mechanical ventilation (collected from the 2014/15 season), death during hospitalization, and length of hospital stay (LOS)—all considered markers of disease severity). We also captured three International Classification of Diseases Ninth or Tenth Revision (ICD-9 or ICD-10) codes assigned to the patient at discharge. The ICD codes were used to identify complications that were associated with the reason for admission (e.g., viral infection that could have evolved to pneumonia or triggered the worsening of underlying conditions) or occurred during hospitalization (e.g., sepsis). Among the children hospitalized with viral infections, complications involving the respiratory tract such as pneumonia and exacerbations of chronic lung disease as well as outside the respiratory tract such as febrile seizures and encephalopathy have been reported. Therefore, we defined complication as any diagnosis of an acute disease process such as pneumonia or bacteremia/sepsis that was grouped in broad categories (Appendix A). Codes associated with symptoms or signs and laboratory abnormalities were not considered. We also used ICD codes to complement information on the presence of comorbidities (Appendix A). Underlying medical conditions included recurrent wheezing/asthma, heart disease, anemia, neurological/neuromuscular diseases, chronic lung disease, chronic renal disease, chronic autoimmune disease, endocrine system disease other than diabetes, diabetes, recent cerebrovascular disease, chronic liver disease, neoplastic disease, and prematurity. Finally, ICD codes were used to identify potential etiologic agents for the condition associated with the hospitalization (Appendix A). We compared them with the results from the multiplex RT-PCR, which was conducted as part of the prospective investigation. The viral multiplex results were not available for patient management nor shared with administrative staff and therefore were not used to guide discharge diagnosis coding. The LOS was recorded as the number of nights spent on the hospital. In case a child was admitted in several wards, the LOS was the sum of the lengths spent in the different wards. We divided age into four categories, namely <6 months, 6 months to <1 year, 1 to <5 years, and 5 to 17 years based on their age at the day of admission.

### 2.5. Statistical Method

Categorical data were described using proportions and compared using Chi-square tests or Fisher tests when the Chi-square hypotheses were not met. Continuous data were compared using t-Student tests or Wilcoxon tests when the conditions of the application of a *t*-Student test were not met. Confidence intervals for single proportions were calculated using the exact binomial method (Clopper–Pearson method). Logistic regression models were used to determine the odds of complications associated with the presence of at least one comorbidity. A *p*-value < 0.05 was indicative of statistical significance. All analyses were performed using R software.

## 3. Results

### 3.1. Demographic Characteristics and Underlying Health Status of Hospitalized Children

From 8680 eligible patients <18 years admitted to the study hospitals from November 2011 through July 2019, 6069 patients were included in the analysis after applying the exclusion criteria (Figure 1).

Almost half of them were <1 year of age (38.9% were <6 months and 11.1% were 6 months to <1 year), and 56.8% were male. The median age among patients <1 year was 2.4 months (interquartile range [IQR], in months: 1.2–4.8) and was 2 years (IQR, in years: 1.6–4.0) for those aged ≥1 year. A total of 1600 patients (26.4%) reported at least one comorbidity, with the majority having only one comorbidity. Prematurity, lung disease, recurrent wheezing/asthma, heart disease, anemia, and neurological/neuromuscular diseases were reported in 10.5%, 8.8%, 4.7%, 2.5%, 2.1%, and 1.4% of cases, respectively. Other comorbidities were reported in less than 1% of patients (Table 1).

### 3.2. Impact of Comorbidities on the Presence of Complications and Disease Severity

In total, 4909 patients (80.9%) had at least one complication associated with their respiratory illness hospitalization. This proportion was similar between patients with and without comorbidities. Overall and per presence of comorbidity, complications were mostly associated with sinorespiratory diagnoses. Pneumonia (21.9% vs. 18.2%), asthma exacerbation/reactive airways (13.4% vs. 5.8%), pneumothorax (2.6% vs. 1.4%), diabetes decompensation (0.4% vs.0.03%), thrombocytopenia (0.4% vs. 0.1%), and musculoskeletal diagnosis (0.4% vs. 0%) were significantly more frequently reported among those with comorbidity than those without, whereas bronchitis/bronchiolitis (42.4% vs. 36.8%), febrile seizures (6.2% vs. 4.6%), and urinary tract infections (0.9% vs. 0.3%) were more often reported among those otherwise healthy (Figure 2). Overall, the median length of hospitalization was 4 days (IQR: 3–6). Very few patients were admitted to the ICU (n = 78, 1.3%) or required mechanical ventilation (n = 21, 0.6%). No patient died during hospitalization. Children with comorbidities were more often admitted to the ICU (2.3% vs. 1.0%), but there was no significant difference regarding the need for mechanical ventilation (1.0% vs. 0.4%), and the length of stay in the hospital did not differ by presence of comorbidity (Table 2).

### 3.3. Impact of Comorbidities Associated with Selected Viral Pathogens

In total, 46% of patients (n = 2790) had no respiratory virus detected by RT-PCR (based on the viral panel used, Table 3), with a slightly higher proportion of negatives among patients without comorbidities compared to those with (50.0% vs. 44.5%, Appendix A). In general, the most frequently detected viruses were RSV (23.1%), followed by rhinovirus/enterovirus (9.5%) and influenza (7.2%). RSV was detected in a slightly higher proportion among patients without comorbidity than those with (24.3% vs. 19.8%, respectively, Appendix A), but there was no other pattern observed among virus-specific detection by comorbidity. Codetection corresponded to 4.6% of the patients and was more frequent in patients without comorbidity than in the patients with comorbidities (5.1% vs. 3.3%, Appendix A). The percentage of codetection was 5.2% among those <6 months, 6.7% among 6–11 months, 4.7% among 1–4 years, and 0.4% among those 5–17 years (Appendix A). Most of the codetections observed were between RSV and rhinovirus/enterovirus, RSV and human seasonal coronavirus, and RSV and bocavirus, with few detections including more than two different viruses (Appendix A). RSV was the most frequently detected virus among those <1 year of age whereas rhinovirus/enterovirus and influenza were the most frequent viral pathogen among school-aged children (i.e., 5 to 17 years—Figure 3 and Appendix A).

Complications by virus varied substantially, as can be seen in Figure 4, with laboratory-confirmed RSV patients mostly presenting with bronchitis/bronchiolitis (74%). Bronchitis/bronchiolitis was also present among cases with human metapneumovirus, parainfluenza, and bocavirus (>40% of cases) and in >50% of those with codetection. The most frequent complications identified among children with laboratory-confirmed influenza were pneumonia (18.3%), febrile seizures (13.7%), and bronchitis/bronchiolitis (11%). Among patients with no respiratory viruses detected by RT-PCR, the most commonly identified complications were also bronchitis/bronchiolitis (30%) and pneumonia (22.1%), with no differences by presence of comorbidity (Appendix A).

Based on ICD discharge diagnoses, 25.8% ([6069 − 4505]/6069) of patients had a pathogen-specific code (Table 3). The most frequent pathogens coded were RSV (18%) and influenza (4%). No difference was observed between the two groups of patients with and without comorbidities, except for RSV, which was coded in a higher proportion in patients without comorbidity than in patients with comorbidities (19.6% vs. 13.6%). There was a small number of other specific etiologies coded at discharge associated with a variety of pathogens including pneumococcus pneumonia and meningitis, Hemophilus influenzae, rotavirus, and measles (Appendix A). Overall, 58.9% of RSV and 75.2% of influenza ICD coded cases were confirmed by multiplex RT-PCR conducted as part of our prospective study. Almost 22% ([1402 − 1093]/1402) of laboratory-confirmed RSV cases were missed when considering the ICD codes only. For influenza, 42.3% ([437 − 246]/437) of the laboratory-confirmed influenza cases were missed when considering the ICD codes used. Approximately 20% of patients with ICD codes for Streptococcal pneumoniae and Hemophilus influenzae had multiplex RT-PCR positive for RSV (Table 3).

## 4. Discussion

In our dataset, most (~75%) of the respiratory-associated hospitalizations were among otherwise healthy children. Nonetheless, children with comorbidities were twice as likely to be admitted to the ICU, even though we did not find differences in other severity markers like mechanical ventilation, death, or LOS by presence of comorbidity. Overall, 80% of children had complications associated with their respiratory illness, and the distribution of certain complications varied by presence of comorbidity. For instance, pneumonia and pneumothorax were more often diagnosed among children with comorbidity. Asthma exacerbation/reactive airways and diabetes decompensation are also a reminder that respiratory illness can trigger the worsening of chronic conditions in children. Nonetheless, our findings corroborate the healthcare burden associated with these infections among children who were otherwise healthy, suggesting that preventive interventions targeting only children with comorbidities would have a lesser impact on reducing healthcare utilization associated with respiratory illnesses.

The role of respiratory viruses on child morbidity and mortality is well-documented [1,2,27,28,29]. More recently, considering the frequency of respiratory viral cocirculation, the association between disease severity and the codetection of respiratory pathogens have been further explored. A study among children <15 years hospitalized with acute respiratory illness described those with viral codetection being more than five times as likely to be admitted to the ICU [30]. Codetection has also been identified as a risk factor for complications among children <5 years hospitalized with COVID-19, where a higher frequency of oxygen support and ICU admissions were described, even after adjusting for the presence of comorbidities [31]. Another study that focused on hospitalized infants with RSV described frequent pathogen codetection with rhinovirus, seasonal coronavirus, Streptococcus, and Hemophilus species, emphasizing the strong contribution of codetection, with Hemophilus being associated with higher overall severity (based on their own severity score) [32]. We did not find any association between codetection and severity in our analysis, which could be due to the small sample size and/or differences in the underlying characteristics of the population or access to care. Moreover, data on pathogens derived from ICD codes are less reliable than medical notes and the direct ascertainment of laboratory test results. The availability of viral diagnostic tools to improve the identification of respiratory illness etiology could become routine care, as this may contribute to improving infection control recommendations, the management of at risk patients, and guide patient care related to therapeutics (antivirals, monoclonal, antibiotics therapy). This may also help reduce pediatric antibiotic use and prevent the spread of antibiotic resistance, especially if these interventions are targeted in children with chronic conditions who may be more likely to be prescribed antibiotics [33,34,35].

Exploring other viral etiologies may be important to further understand their contribution to the burden of respiratory hospitalizations in pediatric populations. For instance, human parechovirus has been described as an important cause of disease in neonates and young children and can lead to fever, sepsis-like syndrome, and/or neurologic illness (seizures, meningitis), which can be difficult to distinguish from other respiratory infections in very young infants [36]. Enterovirus D68 (EVD-68) has also been associated with increased hospitalizations among young children presenting with severe bronchiolitis [37]. EVD-68 targets are not in commercial laboratory tests and were not part of our multiplex. Nonetheless, children with EVD-68 are often detected among those testing positive for rhinovirus/enterovirus, which were the second most detected viruses in our study population, even though our surveillance period focused on the colder months of the year and likely missed rhinovirus/enterovirus circulating during the spring or summer [38,39]. The European Center for Disease Prevention and Control (ECDC) and the World Health Organization (WHO) have recently emphasized the importance of integrated surveillance for influenza, RSV, COVID-19, and potentially other respiratory viruses [40], with year-round monitoring that could inform clinical and public health practices as point-of-care testing may be resource intense.

Influenza is prevented by vaccination in those as young as 6 months of age or through the vaccination of pregnant people to protect those <6 months [41,42]. However, the use of influenza vaccines is limited among young children in many countries, including Spain where policy currently only targets those with comorbidities and the elderly [43]. Our data showed that many children hospitalized with respiratory illness and related complications were otherwise healthy. Studies have suggested that the prevention of influenza through the vaccination of children can reduce influenza-associated hospitalizations in older adults [44], and the spread of influenza in the community [45,46], potentially reducing overall healthcare utilization.

As described in other studies in high income countries [9,47,48], we found RSV to be the most commonly detected viral pathogen in our young children population, being associated with almost a quarter of hospitalizations. The majority of laboratory-confirmed RSV patients had no underlying condition, and the median age of hospitalizations was 2 months. Hospitalization represents just a fraction of all RSV-related healthcare visits and the burden of RSV on primary care is large [47,48]. Long-acting monoclonal antibodies and/or mother/baby vaccination have been approved for use in Europe and the USA and can provide protection from RSV complications including hospitalizations [49,50,51,52,53,54], aiding in reducing the healthcare burden associated with RSV, as influenza and COVID-19 vaccines are already available tools [55].

Our study had several limitations. Despite the large dataset, severe outcomes or some underlying conditions were rare among this pediatric population, reducing our analytical power. Each year, the period of surveillance was defined around the influenza season, which differ from other respiratory viruses, with variations from year to year. We may have underestimated the contribution of some respiratory viruses because we did not document the year-round circulation and because the multiplex RT-PCR we used had limited viral targets. Moreover, the identification of complications among children hospitalized in our study relied on ICD-9 and ICD-10 codes taken from medical records. Some complications may have been missed, since only three codes were collected per patient. The ICD code groupings developed for this analysis have not been previously validated and may misclassify diseases or outcomes, as we documented when comparing coded pathogens from medical records with the multiplex RT-PCR results. Nonetheless, similar analytical approaches to explore complications have been previously published [56,57].

## 5. Conclusions

In conclusion, our study confirms that comorbidities in children with respiratory illnesses significantly increase the risk of complications and are associated with higher ICU admission rates. However, healthcare utilization is largely driven by otherwise healthy children who can also develop severe complications requiring hospitalization. The contribution of respiratory viruses to healthcare surges during the cold months should be acknowledged when planning interventions for this population. Cost–benefit analyses, considering current vaccination strategies and the potential impact of new respiratory-targeted vaccines, are warranted. Additionally, increased use of point-of-care diagnostic tests and expanded surveillance for respiratory viruses could inform new policies for the prevention and management of pediatric respiratory hospitalizations as well as support research and the development of new vaccines in this field.

## Figures and Tables

**Figure 1 viruses-16-01519-f001:**
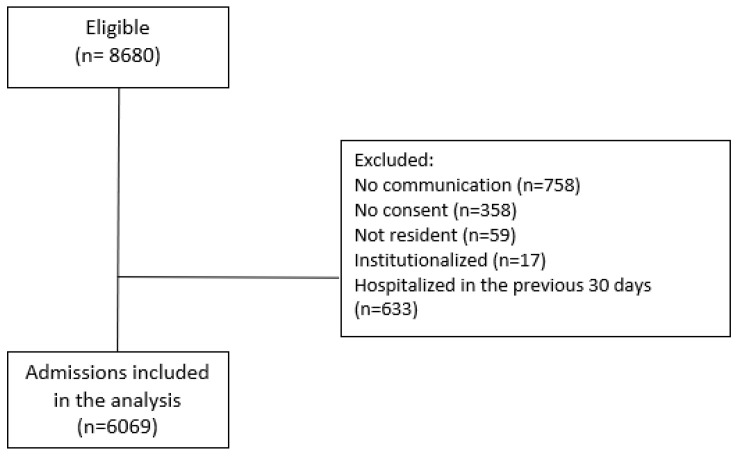
Patient selection process.

**Figure 2 viruses-16-01519-f002:**
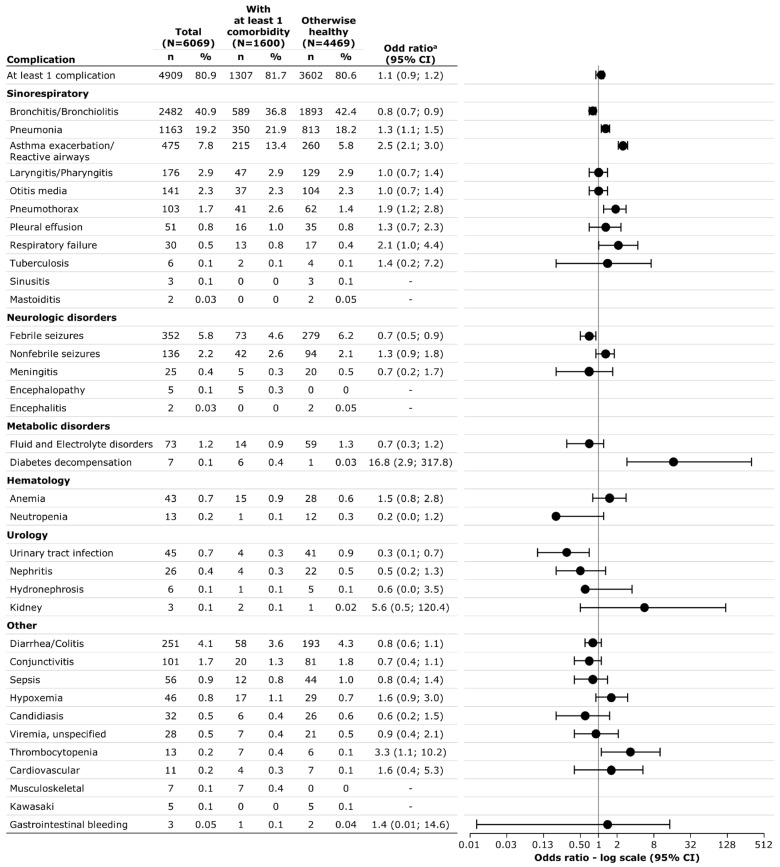
Complications by presence of comorbidity. ^a^ Odd ratio for complication when presenting at least one underlying conditions at admission (otherwise healthy as reference). Abbreviation: CI, confidence interval.

**Figure 3 viruses-16-01519-f003:**
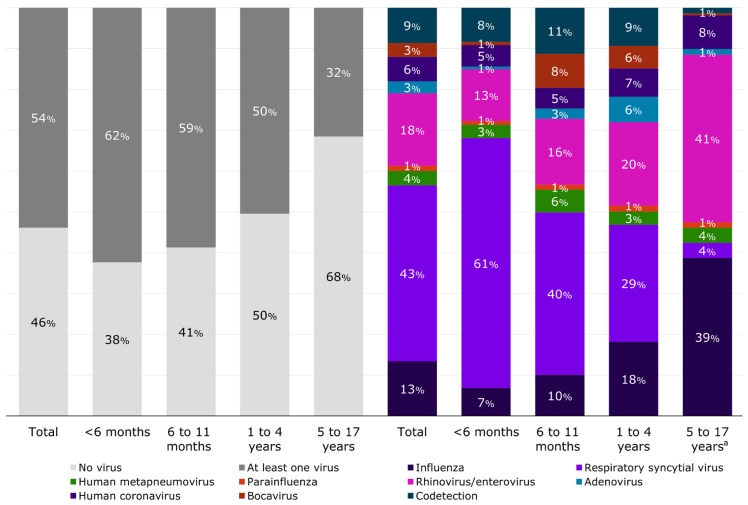
Distribution of respiratory viral pathogens detected by RT-PCR by age group. ^a^ Bocavirus represents 0.5% of the detected viruses in this age group. Influenza includes influenza A(H1N1)pdm09, influenza A(H3N2), influenza B/Yamagata-lineage, influenza B/Victoria-lineage, and influenza not subtyped or with no lineage.

**Figure 4 viruses-16-01519-f004:**
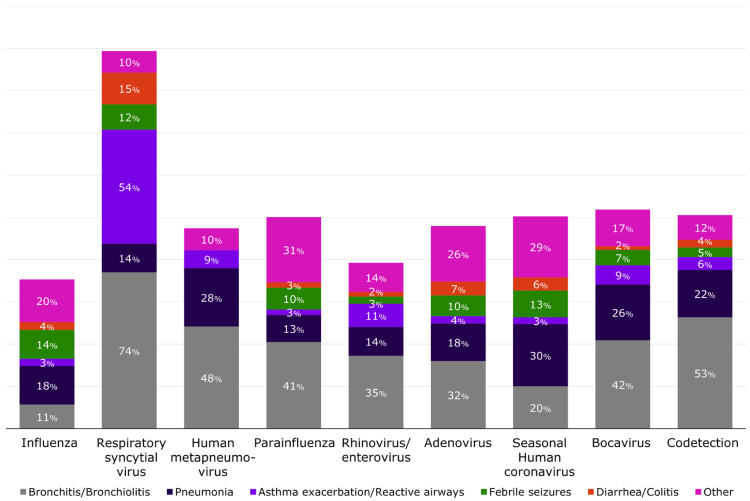
Distribution of selected complications by respiratory viral pathogens detected by RT-PCR, Valencia, Spain 2011/12 through 2018/19. Complications were not mutually exclusive, and the same patient may have developed more than one complication. Influenza includes influenza A(H1N1)pdm09, influenza A(H3N2), influenza B/Yamagata-lineage, influenza B/Victoria-lineage, and influenza not subtyped or with no lineage.

**Table 1 viruses-16-01519-t001:** General characteristics of children at admission, Valencia region, Spain, 2011/12–2018/19.

	Total (N = 6069)	
Characteristic	n	%
Age group ^a^		
<6 months of age	2363	38.9
6 to 11 months of age	677	11.1
1 to 4 years of age	2339	38.5
5 to 17 years of age	690	11.4
Sex		
Male	3447	56.8
Female	2622	43.2
Ratio Male/Female	1.3	
Number of comorbidities at admission		
None	4469	73.6
At least one	1600	26.4
One	1315	21.7
Two or more	285	4.7
Comorbidities		
Prematurity	634	10.5
Lung disease	535	8.8
Recurrent wheezing/asthma	283	4.7
Heart disease	155	2.5
Anemia	125	2.1
Neurological/neuromuscular diseases	84	1.4
Chronic renal disease	53	0.9
Chronic autoimmune disease	24	0.4
Endocrine system disease other than diabetes	21	0.4
Diabetes	17	0.3
Chronic liver disease	9	0.2
Neoplastic disease	8	0.1

^a^ Overall median (interquartile range) age in years was 0.9 (0.2–2.0).

**Table 2 viruses-16-01519-t002:** Severity of patients by presence of comorbidities.

	Total (N = 6069)		Otherwise Healthy (N = 4469)		With at Least 1 Comorbidity (N = 1600)	
Severity	n	% (95% CI)	n	% (95% CI)	n	% (95% CI)
Intensive care admission	78	1.3 (1.0; 1.6)	44	1.0 (0.7; 1.3)	34	2.3 (1.5; 3.0)
Mechanical ventilation ^a^	21	0.6 (0.3; 0.9)	12	0.4 (0.2; 0.7)	9	1.0 (0.5; 1.9)
Length of hospital stay (in days)						
Median	4		4		4	
Q1; Q3	3; 6		3; 6		2; 6	

^a^ Two patients had extracorporeal membrane oxygenation; these were included in mechanical ventilation for the analysis. Data on mechanical ventilation started to be collected during the 2014/2015 season. Abbreviations: CI, confidence interval; Q, quartile.

**Table 3 viruses-16-01519-t003:** RT-PCR results compared to most commonly identified pathogens by ICD code.

	Total ^a^ (N = 6069)	Pathogens Identified through ICD Codes as Part of Patient’s Discharge									
		RSV (N = 1093)	Influenza (N = 246)	Rotavirus (N = 66)	Streptococcus Pneumonia (N = 40)	Hemophilus Influenzae (N = 34)	Epstein-Barr Virus (N = 22)	*Escherichia coli* (N = 22)	Streptococcus (N = 17)	Other Pathogens ^b^ (N = 68)	No Pathogens (N = 4505)
RT-PCR Viral Panel	n (%)	n (%)	n (%)	n (%)	n (%)	n (%)	n (%)	n (%)	n (%)	n (%)	n (%)
No virus detected	2790 (46.0)	295 (27.0)	37 (15.0)	53 (80.3)	27 (67.5)	14 (41.2)	15 (68.2)	16 (72.6)	10 (58.8)	47 (69.1)	2290 (50.8)
Influenza ^c^	437 (7.2)	8 (0.7)	185 (75.2)	1 (1.5)	0 (0)	4 (11.8)	4 (18.2)	0 (0)	2 (11.8)	4 (5.9)	239 (5.3)
Respiratory syncytial virus	1402 (23.1)	644 (58.9)	4 (1.6)	0 (0)	9 (22.5)	8 (23.5)	1 (4.5)	0 (0)	2 (11.8)	6 (8.8)	742 (16.5)
Human metapneumovirus	116 (1.9)	3 (0.3)	2 (0.8)	0 (0)	0 (0)	0 (0)	0 (0)	0 (0)	0 (0)	0 (0)	111 (2.5)
Parainfluenza	39 (0.6)	0 (0)	0 (0)	1 (1.5)	0 (0)	1 (2.9)	0 (0)	0 (0)	0 (0)	2 (2.9)	36 (0.8)
Rhinovirus/enterovirus	579 (9.5)	28 (2.6)	6 (2.4)	1 (1.5)	3 (7.5)	2 (5.9)	1 (4.5)	0 (0)	0 (0)	2 (2.9)	536 (11.9)
Adenovirus	95 (1.6)	5 (0.5)	0 (0)	5 (7.6)	1 (2.5)	0 (0)	0 (0)	1 (4.5)	0 (0)	1 (1.5)	86 (1.9)
Human coronavirus	197 (3.2)	18 (1.6)	0 (0)	0 (0)	0 (0)	2 (5.9)	0 (0)	3 (13.6)	1 (5.9	4 (5.9)	165 (3.7)
Bocavirus	110 (1.8)	9 (0.8)	3 (1.2)	2 (3)	0 (0)	1 (2.9)	0 (0)	1 (4.5)	1 (5.9)	0 (0)	95 (2.1)
Codetection	281 (4.6)	78 (7.1)	9 (3.7)	0 (0)	0 (0)	2 (5.9)	1 (4.5)	0 (0)	1 (5.9)	2 (2.9)	188 (4.2)

^a^ The columns do not add up to the total number reported here as a same patient may have had none or more than one pathogen identified in the ICD codes. Additionally, the rows do not add up to the total numbers reported in the heading of the table because of 23 underdetermined samples. ^b^ Other pathogens are listed in Appendix A. ^c^ Includes influenza A(H1N1)pdm09, influenza A(H3N2), influenza B/Yamagata-lineage, influenza B/Victoria-lineage, and influenza not subtyped or with no lineage.

## Data Availability

The data that support the findings of this study are available from the authors upon reasonable request and with the permission of FISABIO-Public Health. Please contact Ainara Mira-Iglesias (ainara.mira@fisabio.es).

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
