# Peer review of "Pediatric Respiratory Hospitalizations in the Pre-COVID-19 Era: The Contribution of Viral Pathogens and Comorbidities to Clinical Outcomes, Valencia, Spain"

_viruses, 2024, doi:10.3390/v16101519_

Round 1

Reviewer 1 Report

Comments and Suggestions for Authors

The manuscript by Castells and co-workers is a comprehensive description of respiratory illnesses in children in Spain during two pre-COVID epochs. Particularly interesting are the virological results documenting the specific pathogens identified in these hospitalized children. Major points are the preponderance of RSV infections - at least among those where the pathogen was detected - and the importance of co-morbidities in driving seriousness.

The manuscript could benefit from two modifications a minor one and a major one. 

1. The minor modification is some clarification of the proportion of patients included in the study among the many who were hospitalized. If this is in the manuscript, I did not see it. Many clinical investigators help the reader follow the study by the use of a flow chart, and the authors should consider that, including percent of isolations, etc. 

2. Although ranges are provided for many data points in the tables, there is a surprising lack of statistical support for the points made. The authors should consult a statistician for many of the points made, beginning with Table 2 and Fig 1, among others. 

Author Response

Comment 1. The minor modification is some clarification of the proportion of patients included in the study among the many who were hospitalized. If this is in the manuscript, I did not see it. Many clinical investigators help the reader follow the study by the use of a flow chart, and the authors should consider that, including percent of isolations, etc. 

Answer 1. Thank you for pointing this out. We agree with this comment. We have added a flow chart as Figure 1 in the section 3.1 Demographics characteristics and underlying health status of hospitalized children.

Comment 2. Although ranges are provided for many data points in the tables, there is a surprising lack of statistical support for the points made. The authors should consult a statistician for many of the points made, beginning with Table 2 and Fig 1, among others. 

Answer 2. Thank you for this point. We acknowledge that p-values are not consistently reported in our analysis to avoid issues related to test multiplicity. In fact, many journals are now recommending in their new guidelines that “many aspects of the reporting of studies in the Journal, including a requirement to replace P values with estimates of effects or association and 95% confidence intervals when neither the protocol nor the statistical analysis plan has specified methods used to adjust for multiplicity.” [New Guidelines for Statistical Reporting in the Journal | New England Journal of Medicine (nejm.org)]

As recommended by many current statisticians and journals, However, we provide main point estimates with 95% confidence intervals (CI) to indicate the range of values that we expect your estimate to fall between a certain range. In the case of Figure 1, these intervals also show whether the odds ratios are statistically significant (observing overlapping ranges). The significance of differences between groups was primarily based on clinical judgment, though we included p-values when relevant (e.g., supplementary Table 5). We indeed have statistician weighing in the analysis and, with due respect, we would like to continue as is.

Reviewer 2 Report

Comments and Suggestions for Authors

Author Response

Comment 1. Line 36. Spell out RSV on first use of acronym.

Answer 1. Thank you for pointing this out. It has been corrected in the new version of the manuscript.

Comment 2. Line 92-95. Authors should note if these were commercial or in-house developed rRT-PCR assays, and if commercial, include the kit/company name. How one demarcates a positive test result based on Ct cutoff value(s) is an important consideration and therefore should be stated.

Answer 2. Thank you for this comment. We have clarified in the text that the RT-PCR assay used in our analysis is an in-house PCR. Additionally, we have included information on the cycle threshold cut-offs for determining a positive result.

Comment 3. Line 197. Change “viruses” to “virus”.

Answer 3. Thank you for pointing this out. It has been corrected in the new version of the manuscript.

Comment 4. Line 270. Change “Another studies …” to “Another study

Thank you for pointing this out. It has been corrected in the new version of the manuscript.

Round 2

Reviewer 1 Report

Comments and Suggestions for Authors

The authors have responded with some wording that indicates that they do not trust P values. I agree; I did not request p values, but rather that a statistician approve of the associations made, which are now said to be based on their estimation. While the authors accurately quote the NEJM guidelines regarding the problems with p values, further down in those guidelines it is said:

 A large array of methods to adjust for multiple comparisons is available and can be used to control the type I error probability in an analysis when specified in the design of a study.6,7

Since the authors indicate they have consulted a statistician, perhaps they can so indicate in a note or by authorship, naming the individual who is responsible for the accuracy of the methodology used. As I am not a statistician, I will not argue with an appropriate professional. 

Author Response

[comment] The authors have responded with some wording that indicates that they do not trust P values. I agree; I did not request p values, but rather that a statistician approve of the associations made, which are now said to be based on their estimation. While the authors accurately quote the NEJM guidelines regarding the problems with p values, further down in those guidelines it is said:

 A large array of methods to adjust for multiple comparisons is available and can be used to control the type I error probability in an analysis when specified in the design of a study.6,7

Since the authors indicate they have consulted a statistician, perhaps they can so indicate in a note or by authorship, naming the individual who is responsible for the accuracy of the methodology used. As I am not a statistician, I will not argue with an appropriate professional. 

[answer] Thank you for your comment. To clarify, it is not about the authors ‘trusting or not’ p-value. If we needed to estimate p-values beyond 95% CI, we would. The first author is a statistician with over 20 years of experience and the co-authors are senior epidemiologists with extensive publication track. The authors are confident of the analysis, this is a descriptive observational study, and the conclusion of the paper does not overstate the findings. We acknowledge that this is a descriptive paper, and our recommendation is to leave it as is. I hope we can settle this situation.